# Are Health Literacy and Physical Literacy Independent Concepts? A Gender-Stratified Analysis in Medical School Students from Croatia

**DOI:** 10.3390/children9081231

**Published:** 2022-08-15

**Authors:** Marijana Geets Kesic, Mia Peric, Barbara Gilic, Marko Manojlovic, Patrik Drid, Toni Modric, Zeljka Znidaric, Natasa Zenic, Aleksander Pajtler

**Affiliations:** 1Faculty of Kinesiology, University of Split, 21000 Split, Croatia; 2Faculty of Sport and Physical Education, University of Novi Sad, 21000 Novi Sad, Serbia; 3Faculty of Civil Engineering, Transportation Engineering and Architecture, University of Maribor, 2000 Maribor, Slovenia

**Keywords:** health behaviors, knowledge translation, public health, community-engaged research, youth, body build

## Abstract

Health literacy (HL) and physical literacy (PL) are concepts responsible for achieving and maintaining positive health behaviors. This study aimed to investigate gender-specific associations: (i) between PL and HL; and (ii) among HL, PL, and body composition. We observed 253 students attending health-area high schools from southern Croatia (181 girls; 16.9 ± 1.4 years). HL was assessed by the European Health Literacy Survey Questionnaire, PL by the PLAYself questionnaire, and body composition by bioimpedance analysis. The *t*-test was used to assess the differences between genders, and Pearson’s correlation coefficients were calculated to establish the associations between variables. The results showed a similar level of HL (*t*-test = 0.2; *p* = 0.83) and PL (*t*-test = 0.01; *p* = 0.99) in boys and girls. Significant but small correlations were identified between HL and PL only in the girls (<10% of common variance). The body composition indices were significantly correlated with PL only in the boys (15–20% of common variance). Our research highlights the necessity of the independent evaluation of HL and PL in adolescence. Further studies evaluating other indices of health status in relation to PL and HL are warranted.

## 1. Introduction

From the perspective of public health, adolescence is one of the most important stages of life, because in this period individuals shape their own health behavior, as parental control decreases and the adolescent’s autonomy increases [1]. In other words, adolescents make health decisions that will determine their health outcomes in later stages of life. Among the most important factors of health behavior in this period of life are nutritional habits, the level of physical activity, and substance misuse (i.e., alcohol and tobacco consumption). Indeed, education regarding physical activity and healthy nutrition begins from a very young age (i.e., as a child) in the perspective of active games [2]. This emphasizes the importance of early education and the acquisition of healthy habits, related to improved physical fitness and cardio-metabolic health [3].

Worryingly, more than 80% of children and adolescents do not have sufficient physical activity levels [4]. Globally, the prevalence of tobacco and alcohol consumption has increased [5], and youth have poor nutritional habits [6]. All of the factors mentioned above are considered leading causes of chronic noncommunicable diseases and could be prevented with adequate health-promoting strategies [7]. Thus, it is crucial to identify the determinants and concepts that influence the formation of adolescents’ health behaviors to prevent adverse health outcomes. Several concepts and skills are deemed important in order for adolescents to adopt, nurture, and maintain healthy behaviors, including health literacy and physical literacy [8,9]. 

Health literacy (HL) is defined as “the characteristics and social resources needed for people to access, understand and use information to make decisions about health. HL includes the capacity to communicate, assert and enact these decisions” [10]. Therefore, HL has been accepted as an effective concept for health promotion [11]. Indeed, in a comprehensive review, it was shown that the individuals with lower HL had poorer global health status, were more likely to express symptoms of depression, and had higher all-cause mortality rates [8]. In addition, a study on Indonesian adolescents identified that HL was related to health behaviors, including physical activity [12], while another study recorded that HL was associated with health-related quality of life (i.e., self-perception of mental and physical health conditions) among Chinese schoolchildren [13]. Moreover, low levels of HL were linked to increased body weight and obesity [14,15] and poor nutritional habits, including increased sugar, fat, and salt intake in children and adolescents [16].

Another important concept for achieving positive health behaviors and outcomes is physical literacy (PL). The most commonly used definition of PL is: “the motivation, confidence, physical competence, knowledge, and understanding to value and take responsibility for engagement in physical activities for life” [17]. Thus, it has been suggested that PL should be considered a determinant of health, as it leads to increased physical activity resulting in improved physical, social, and mental health [18]. Indeed, numerous studies reported a positive relationship between PL and health indicators [9,18,19]. A review study by Cornish et al. (2020) reported that PL was associated with numerous health indicators, including body mass index, waist circumference, body weight, cardiorespiratory fitness, systolic blood pressure, health-related quality of life, and physical activity [9]. Moreover, in Spanish children, body composition was correlated to PL, with BMI, fat mass, and percentage of fat mass being inversely related to lower PL scores [19]. Similarly, a study on Canadian children and adolescents found a negative relationship between PL and the percentage of fat mass [20].

Based on Bandura’s social cognitive theory, the determinants of one’s behaviors include self-efficacy (defined as an individual’s belief in their capability to organize and carry out actions to reach results), attitude, knowledge, and social support [21,22]. From the previous brief descriptions of HL and PL, it can be theorized that HL and PL are associated with the social cognitive determinants of health behavior, with an emphasis on self-efficacy and knowledge, and could be responsible for influencing health outcomes. However, the interrelationships between HL and PL are rarely reported, while there is an evident lack of knowledge on the associations that may exist among HL, PL, and body composition as important determinants of health status in adolescence. The main aim of the study was to evaluate the gender-specific associations between HL and PL in a sample of Croatian high school students. Additionally, we examined gender-specific associations between PL, HL, and body composition. Initially, we hypothesized that HL and PL would not be significantly correlated in gender-stratified analyses. As a methodological remark, we must note that the previous studies on adolescents regularly confirmed differences between genders in PL, HL, and body composition [23,24,25,26]. Therefore, we considered a gender-stratified analyses as being more appropriate in studying the relationships among PL, HL, and body composition. Otherwise, the difference between the genders could influence the results of the correlation analyses (i.e., gender could be a covariate), resulting in inappropriate and ecologically non-valid findings/conclusions.

## 2. Materials and Methods

### 2.1. Participants and Design of the Study

In this cross-sectional study, the participants were 253 adolescents (16.9 ± 1.4 years of age, 181 females) attending health-area high schools in Split-Dalmatia County, in southern Croatia. On the basis of a correlation between HL and PL of 0.30, established in a pilot study on Croatian college students [27], with a type-I-error rate of 0.05, and a type-II-error rate of 0.20, the necessary sample size was 85 participants [28]. The inclusion criterion for the study was that the mean age of the participants fell within the World Health Organization’s (WHO) definition of adolescents (10–19 years of age). The exclusion criterion was students with acute inflammatory disease (e.g., COVID-19). 

The study was approved by the Ethical Board of the University of Split, Faculty of Kinesiology (EBO: 2181-205-02-01-21-0011; date of approval, 23 September 2021). After ethical approval, one of the first authors of the study presented the aim and procedure of the study to all of the school classes. The written informed consent was signed by the interested student, or by a parent/legal guardian (for those younger than 18 years), prior to the study’s initiation. Therefore, 400 consent waivers were distributed. A total of 295 waivers were recovered, with a response rate of 74%. The participants were aware that they could withdraw from the study at any time. In total, 42 participants withdrew, which resulted in a final sample of 253 participants. The measurements were conducted in April and May 2022, during the school day from 08:00 am to 10:00 am.

The study variables and idea of the investigation are presented in Figure 1.

### 2.2. Variables and Measurement

The variables included evaluation of HL, PL, and anthropometric/body-build indices.

The HL was evaluated using the European Health Literacy Survey Questionnaire (HLS-EU-Q), developed by Sorensen et al. [29]. The HLS-EU-Q comprises 47 questions, which measure an individual’s ability to access, understand, appraise, and apply health-related information [30]. The general index of HL was constructed using a 4-point Likert scale, with responses from very difficult−1 to very easy−4. The score was calculated using the following formula: index = (mean − 1) × (50/3). The HL scale (from 0 to 50) was formed, where 0 represented the lowest and 50 the highest score. Four levels of HL were defined: inadequate (from 0 to 25); problematic (26–33); sufficient (34–42); and excellent (43–50). The HLS-EU-Q was first translated from English into Croatian and then back-translated by two different professional translators. Geets-Kesic et al. (2022) conducted the survey among 134 Croatian students and demonstrated that the HLS-EU-Q had good reliability (Kappa:0.79, with 91% of the answers equally responded to) [27]. The HLS-EU-Q was evaluated using the online platform SurveyMonkey (SurveyMonkey Inc., San Mateo, CA, USA).

The PL was evaluated by the PLAYself questionnaire. It is a self-assessment questionnaire designed to evaluate the current degree of PL. Four groups of questions measure (i) the affective and cognitive aspect of PL; (ii) the environmental ability; (iii) the estimation of literacy, numeracy, and physical literacy in different settings; and iv) fitness [31]. The final score was the sum of the first three groups of questions divided by the number of questions. A total score of 100 points indicated the highest self-perceived PL. In this study, we used the Croatian version of the PLAYself questionnaire, which was previously shown to be reliable and valid among Croatian adolescents [32,33]. PLAYself was conducted using the online platform SurveyMonkey (SurveyMonkey Inc., San Mateo, CA, USA).

The anthropometric/body-build indices included measurement of body mass (in 0.1 kg), body height (in cm), fat free mass (in 0.1 kg), visceral fat (level), and muscle mass (in 0.1 kg), while the body mass index (BMI = mass (kg)/height^2^(m)) was also calculated. The body composition was measured by the bioimpedance scale (Tanita BC 418 scale; serial number: 15010067, 2015, Tanita, Tokyo, Japan). The measurement was completed in the school laboratory by a medical doctor—the first author of the study (privacy was secured) at room temperature in the morning from 08:00 to 10:00. Prior to the measurement, the procedure and protocol were explained to each student. During the measurement, students were dressed in their underwear, and they were barefoot.

### 2.3. Statistical Analysis

The normality of the distributions was checked by the Kolmogorov–Smirnov test of normality. All of the variables were evidenced as being normally distributed, and the means and standard deviations were reported.

The *t*-test for independent samples was used to determine the possible differences between the boys and girls.

Pearson’s product moment correlation coefficient, as a measure of correlation between normally distributed variables, was calculated to evaluate the association between the pairs of variables, and this was performed for the total sample, and was gender-stratified.

Statistica 13.5 (Tibco Inc, Palo Alto, CA, USA) was used for all of the calculations, and *p*-value of 0.05 was applied.

## 3. Results

The descriptive statistics for the measured variables and differences between the genders are presented in Table 1. In brief, the boys and girls differed significantly in the anthropometric-body composition measures; the boys were taller, heavier, had a lower level of body fat, and more muscle mass than the girls. Generally, there were no significant differences between the genders in HL and PL, except for the PLAY literacy sub-score where the girls achieved higher results than the boys.

The correlations between the study variables for the total sample of participants are presented in Appendix A.

For the boys, the correlations between variables are presented in Table 2. Apart from significant correlations between the anthropometric/body composition variables (moderate to strong correlations between the body composition indices and body mass; >35% of the common variance), and the significant correlations among the PLAYself sub-scores and the PLAYself total score (25–82% of the common variance), the associations between anthropometric/body composition with PL were small. Specifically, the indices of adipose body mass were significantly correlated to PL, with 16–20% of the common variance. Finally, the anthropometric/body composition indices were not significantly correlated to HL for the boys.

The correlation between HLS-EU-Q and PLAYself among the boys is presented in Figure 2. The correlation did not reach statistical significance (Pearson’s r = 0.08; *p* > 0.80).

In the girls, the anthropometric/body composition indices were intercorrelated (>40% of the common variance). Meanwhile, no significant correlation between anthropometrics/body composition, PL and HL was evidenced (<2% of the common variance). Three of the PL sub-scores were significantly correlated to HL (7–13% of the common variance), indicating significant associations between the PLAYself _numeracy_, PLAYself _literacy_, and PLAYself _physical literacy,_ with HL (Table 3).

When the PLAYself and HLS-EU-Q results were correlated for the girls, the variables shared 10% of the common variance (Pearson’s r = 0.31; *p* < 0.001), indicating a small but significant association between HL and PL among the studied adolescent girls (Figure 3).

## 4. Discussion

This study aimed to investigate gender-specific associations between PL, HL, and body composition in Croatian adolescents. There are several main findings of this study: (i) the boys and girls did not differ, either in PL or in HL; (ii) HL and PL had low intercorrelations; and (iii) the body composition indices were not correlated to HL.

### 4.1. Gender Differences in Health Literacy and Physical Literacy

The finding that the boys and girls did not differ in PL is in accordance with previous studies. A study on Canadian children and adolescents aged 8–14 years recorded no differences between males and females in confidence, motivation, and knowledge domains of PL [34], which was also confirmed in a study on Canadian children aged 8–12 years [35]. A very recent study conducted on high school students aged 14–18 years from continental Croatia also did not find differences between the boys and girls in PL scores [32]. The authors explained such findings with the fact that the students have standardized fitness norms in their physical education classes, which enables them to compare their physical level within their gender. More specifically, the adolescents were most likely judging themselves within and not between genders; this is directly supported by a previous study that showed that youth judge themselves within a similar group (age, sex, and ethnicity), which leads to a precise evaluation of their abilities [36]. Collectively, it probably resulted in the nonsignificant difference in PL between genders in our study as well.

However, the lack of differences in the HL between genders is not in accordance with previous global studies. Specifically, a study on Korean adults reported that the females indicate higher HL than males in understanding medical forms and information [24]. Moreover, a study on non-medical college students from Egypt reported that the females had higher levels of HL than males, which was determined to be due to females’ more frequent online health-information seeking [23]. Further, a review study on the gender differences in the mental HL of university students between 16 and 25 years old revealed that females were more able to identify common mental health disorders and had higher HL than males [37]. However, it must be noted that the previous studies that recorded differences in HL between genders investigated very diverse samples and adults, while our study included a significant proportion of students attending medical school, i.e., future health professionals. Therefore, it was logical to expect that all of the medical school students, regardless of gender, would possess high levels of HL, consequently leading to no differences in the HL scores between the genders in our study. In addition, our study was completed in the period of the COVID-19 pandemic, which probably influenced the level of awareness on health issues in all of the participants, even the boys, as recently suggested [38].

### 4.2. Associations between Health Literacy and Physical Literacy

HL and PL were poorly intercorrelated, which could be explained by the fact that HL and PL, although both were related to health behaviors, are actually not the same concepts. In brief, HL relates to making sound and positive health decisions, leading to lifelong, health-promoting behaviors [39]. On the other side, PL promotes engagement in physical activity, which is one of the essential positive life habits that improves health [20,40]. It has been stressed that the terms HL and PL should be used with caution, as those terms are sometimes used synonymously [41]. The most probable reason is that the school subject is often called “Physical and Health Education” (or similar) which creates confusion and leads to considering health literacy and physical literacy as one concept [41]. However, those concepts relate to, and describe, different types of literacy, which is clearly supported by our results.

The lack of association between HL and PL in our study can also be attributed to the measurement tools we used to assess HL and PL. Namely, the HLS-EU-Q consists of items regarding the process of accessing/obtaining, understanding, processing/appraising, and applying/using health information in three domains, including healthcare (i.e., information on clinical or medical issues), disease prevention (information on health risk factors), and health promotion (updating oneself on health determinants in the physical and social environment) [30]. On the other side, PLAYself evaluates PL as a multidimensional construct comprising the competence, confidence, and knowledge to be active, and motivation to use movement skills. Additionally, PLAYself is a self-report measure of PL with items assessing affect (confidence, motivation), knowledge, and understanding of physical activities and movement [31]. Therefore, although both of the questionnaires assess the constructs that relate to health-related behaviors, they are actually distinct concepts and should be assessed separately.

### 4.3. Associations among Body Composition, Health Literacy, and Physical Literacy

The body composition indices did not strongly correlate to PL and HL, which is somewhat surprising, as we know that body composition is an important indicator of overall health status (which is definitively related to HL) and physical fitness (which is known to be strongly associated with PL) [9,42]. Indeed, opposite to the results of our study, several studies recorded significant associations between body composition and anthropometric variables with HL in children and adolescents. A study on U.S. children and adolescents aged 6–19 years reported an inverse relation between HL and body mass index [43]. Moreover, a study on adolescents aged 12–19 years recorded that HL was strongly related to obesity; adolescents with low HL were more likely to be obese [44]. The reason for this was found in the fact that adolescents with low HL have poor nutritional habits, with an increased intake of sugar, fat, and salt [16,45], which leads to increased weight, body mass index, and body fat percentage.

However, not all of the studies have indicated a significant association between HL and anthropometrics/body composition. For example, a study on children aged 8–11 years did not find a significant correlation between HL and body mass index, which was explained by the limited variation in body mass index, as most of the children were within the normal range [46]. We can offer a similar explanation for the lack of association between HL and body-build indices in our research. First, when compared to studies where the authors evidenced significant associations between anthropometrics/body composition and HL, we observed a relatively narrow age span (14–18 years in our research, and 6–19 years and 12–19 years in the studies of our respective colleagues) [43,44]. Second, our adolescents had a mean body mass index score of 22.86 ± 8.31, which falls within the normal range [47]. Simply put statistically, the narrow variance limited the possibility of reaching the higher correlation coefficient and statistical significance of the association [48].

The small correlation between PL and the anthropometric/body composition indices is more complex, especially when we consider that the studies regularly note clear correlations between body composition and PL. A study on Canadian youth evidenced a negative association between PL and body fat percentage [20], meaning that the children with low PL scores had an increased body fat percentage, which is an indicator of increased risk for metabolic dysfunctions and cardiovascular risk factors [49,50]. Moreover, it was shown that the children with healthy (i.e., lower) weight, body mass index, and waist circumference were more likely to have higher PL scores than the children with an unhealthy weight, body mass index, and waist circumference for their age and height [51,52]. However, the studies that recorded associations between PL and body composition mostly investigated a composite PL score, which consisted of physical, behavioral, cognitive, and affective domains [20]. Meanwhile, in this study, we investigated only the cognitive and affective domains of PL. Thus, it could be hypothesized that the cognitive and affective domains of PL do not relate to body composition and anthropometric variables in the studied adolescents. In addition, we must note that the anthropometric/body composition indices were significantly correlated to PL in the boys studied here, but in the total sample, due to a larger number of girls (and the lack of association between anthropometric/body composition and PL among the girls), the overall correlation did not reach statistical significance.

### 4.4. Limitations and Strengths

The main limitation of this study was the cross-sectional nature of the investigation. Therefore, causality cannot be determined. Next, the tool used for assessing PL (i.e., PLAYself) is probably not the best option for assessing PL in high-school adolescents, nor for assessing all of the domains of PL as it includes only the cognitive and affective ones. Furthermore, this study included anthropometric/body-composition variables as the only indicator of health status, and in future studies, other indices of health status should be explored. Finally, HL and PL were evaluated through questionnaires, which raises the possibility of receiving socially desirable and not honest answers. However, this problem was eliminated to some extent, as the participants answered anonymously (they used only codes to enable the researchers to match their answers to other investigated variables).

As far as the authors are aware, this is one of the first studies investigating HL in Croatian adolescents, which is this study’s major strength. Moreover, the studies investigating the associations among PL, HL, and body composition are generally lacking, while there is also an evident lack of investigations where this problem is analyzed through a gender-specific approach. Therefore, we believe that this study will contribute to the overall knowledge in the field and will hopefully initiate further research.

## 5. Conclusions

We found weak associations between HL and PL; our research highlights the necessity for the separate evaluation of each of these important abilities in adolescence.

The significant association between PL and body composition was established for the boys, but not for the girls. Therefore, improvement of PL could be effective for the improvement of body composition for the boys, but not for the girls. However, before drawing final conclusions, additional correlational studies examining the more heterogenous sample of participants are needed.

In future studies, all of the aspects of PL should be observed to objectively evaluate PL itself and to establish the associations that may exist between PL, HL, and health-status. In addition, future studies should focus on the additional indices of health (other than body composition), in order to objectively evaluate the associations that may exist among HL, PL, and health status.

## Figures and Tables

**Figure 1 children-09-01231-f001:**
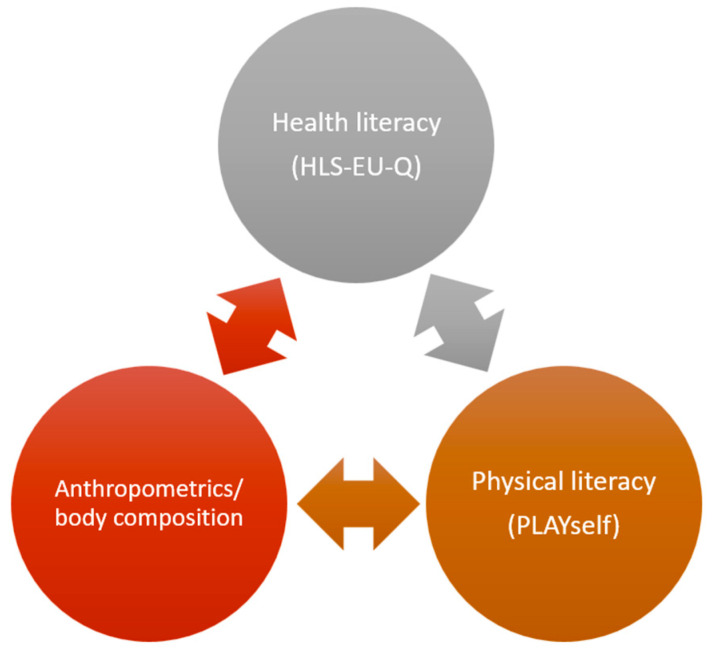
Study variables and studied associations.

**Figure 2 children-09-01231-f002:**
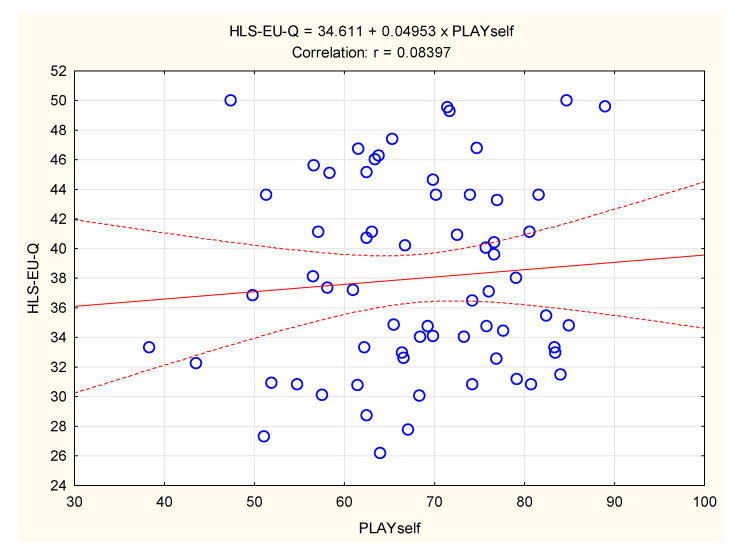
Correlation between physical literacy (PLAYself) and health literacy (HLS-EU-Q) in boys (scattered lines present 95% confidence intervals).

**Figure 3 children-09-01231-f003:**
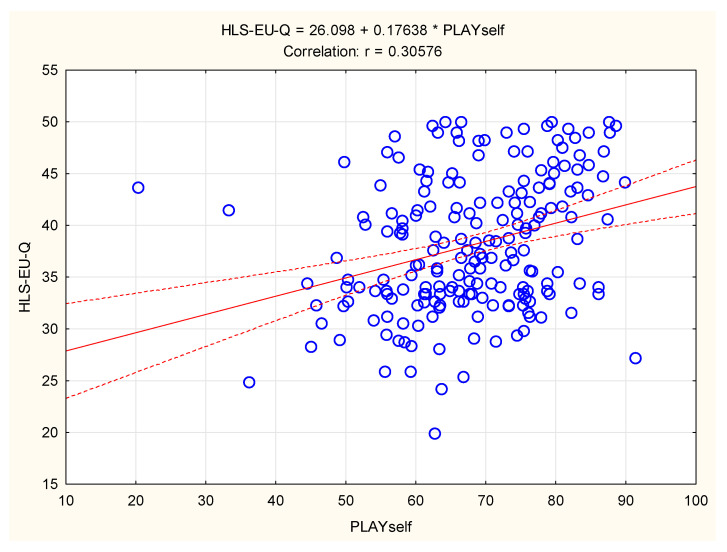
Correlation between physical literacy (PLAYself) and health literacy (HLS-EU-Q) in girls (scattered lines present 95% confidence intervals).

**Table 1 children-09-01231-t001:** Descriptive statistics and differences between genders in study variables.

	Boys (*n* = 68)		Girls (*n* = 198)		*t*-Test	
	Mean	Std. Dev.	Mean	Std. Dev.	*t*-Value	*p*-Value
Age (years)	17.08	1.31	16.96	1.41	0.62	0.53
Body height (cm)	180.97	14.30	169.27	6.53	9.10	0.001
Body mass (kg)	74.40	12.49	64.48	11.15	6.13	0.001
BMI (kg/m^2^)	24.06	15.30	22.44	3.29	1.40	0.16
Fat mass (kg)	12.73	5.59	18.30	7.36	−5.70	0.001
Fat mass (%)	16.61	4.98	27.42	6.39	−12.69	0.001
Free fat mass (kg)	61.68	8.55	46.36	5.32	17.31	0.001
Muscle mass (kg)	58.90	8.11	43.89	4.45	19.04	0.001
Visceral fat (kg)	2.14	1.51	1.81	1.21	1.18	0.24
PLAYself _environment_ (sub-score)	50.86	15.23	51.16	15.43	−0.13	0.89
PLAYself _self-description_ (sub-score)	73.68	14.51	70.24	15.06	1.63	0.10
PLAYself _literacy_ (sub-score)	72.51	16.25	83.11	17.31	−4.39	0.001
PLAYself _numeracy_ (sub-score)	60.06	21.28	61.07	22.70	−0.32	0.75
PLAYself _physical literacy_ (sub-score)	82.59	18.04	86.60	19.45	−1.48	0.14
PLAYself total (score)	68.05	11.04	68.03	11.36	0.01	0.99
HLS- EU-Q (score)	37.87	6.53	38.06	6.54	−0.21	0.83

Legend: PLAY = Physical literacy Assessment of Youth; HLS-EU-Q = European Health Literacy Survey Questionnaire.

**Table 2 children-09-01231-t002:** Correlations between study variables for boys (* indicates coefficients significant at *p* < 0.05).

	1	2	3	4	5	6	7	8	9	10	11	12	13	14
Age (1)	-													
Body height (2)	0.10													
Body mass (3)	0.25 *	0.28												
BMI (4)	0.03	−0.13	0.91 *											
Fat mass kg (5)	0.03	0.03	0.85 *	0.87 *										
Fat mass % (6)	0.09	−0.08	0.65 *	0.71 *	0.95 *									
Free fat mass (7)	0.35 *	0.40 *	0.95 *	0.81 *	0.64 *	0.37								
Muscle mass (8)	0.35	0.40 *	0.95 *	0.81 *	0.63 *	0.37	0.99 *							
Visceral fat (9)	0.30	−0.19	0.73 *	0.83 *	0.84 *	0.76 *	0.55 *	0.56 *						
PLAYself _environment_ (10)	0.02	−0.03	0.30	0.30	0.39 *	0.40 *	0.20	0.20	0.24					
PLAYself _self-description_ (11)	0.04	−0.04	0.52 *	0.54 *	0.59 *	0.55 *	0.40 *	0.41 *	0.57 *	0.62 *				
PLAYself _literacy_ (12)	0.21	−0.07	0.03	0.06	0.09	0.10	−0.01	−0.01	0.13	0.09	0.30			
PLAYself _numeracy_ (13)	0.03	0.21	0.14	0.05	−0.03	−0.12	0.23	0.23	0.00	0.06	0.27	0.64 *		
PLAYself _physical literacy_ (14)	0.14	−0.34	−0.08	0.05	0.03	0.08	−0.14	−0.14	0.15	0.09	0.38	0.34	0.31	
PLAYself total (15)	0.05	−0.06	−0.40 *	−0.43 *	−0.47 *	−0.44 *	0.31	0.31	0.45 *	0.67 *	0.92 *	0.55 *	0.52 *	0.53 *
HLS-EU-Q (16)	0.04	0.19	0.08	−0.02	0.12	0.15	0.05	0.04	0.01	0.29	0.34	−0.15	0.09	0.28

Legend: PLAY = Physical literacy Assessment of Youth; HLS-EU-Q = European Health Literacy Survey Questionnaire.

**Table 3 children-09-01231-t003:** Correlations between study variables for girls (* indicates coefficients significant at *p* < 0.05).

	1	2	3	4	5	6	7	8	9	10	11	12	13	14
Age (1)	-													
Body height (2)	0.04													
Body mass (3)	0.03	0.66 *												
BMI (4)	0.01	0.28 *	0.90 *											
Fat mass kg (5)	0.01	0.56 *	0.97 *	0.92 *										
Fat mass % (6)	0.03	0.48 *	0.89 *	0.87 *	0.97 *									
Free fat mass (7)	0.09	0.59 *	0.73 *	0.59 *	0.61 *	0.49 *								
Muscle mass (8)	0.09	0.73 *	0.89 *	0.71 *	0.74 *	0.59 *	0.81 *							
Visceral fat (9)	−0.17	0.31 *	0.86 *	0.92 *	0.91 *	0.85 *	0.52*	0.62 *						
PLAYself _environment_ (10)	−0.21 *	−0.04	−0.07	−0.06	−0.09	−0.08	0.11	−0.02	−0.14					
PLAYself _self-description_ (11)	0.07	0.00	−0.11	−0.14	−0.16	−0.19	−0.01	−0.03	−0.16	0.41 *				
PLAYself _literacy_ (12)	0.13	0.05	0.15	0.17	0.19	0.21	0.07	0.07	0.16	0.12	0.05			
PLAYself _numeracy_ (13)	0.11	−0.01	−0.01	−0.02	−0.04	−0.07	0.13	0.04	−0.02	−0.05	0.11	0.20		
PLAYself _physical literacy_ (14)	0.03	0.16	0.07	−0.02	0.05	0.02	0.11	0.10	−0.01	0.25 *	0.39 *	0.19	0.43 *	
PLAYself total (15)	−0.04	0.03	−0.05	−0.09	−0.09	−0.12	0.10	0.02	−0.12	0.61 *	0.85 *	0.32 *	0.42 *	0.66 *
HLS- EU-Q (16)	0.19 *	−0.08	−0.01	0.04	0.00	0.05	0.03	−0.06	−0.03	0.05	0.26 *	0.27 *	0.38 *	0.04

Legend: PLAY = Physical literacy Assessment of Youth; HLS-EU-Q = European Health Literacy Survey Questionnaire.

## Data Availability

Data will be provided to all interested parties upon reasonable request.

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
