# Peer review of "Are Health Literacy and Physical Literacy Independent Concepts? A Gender-Stratified Analysis in Medical School Students from Croatia"

_children, 2022, doi:10.3390/children9081231_

Round 1

Reviewer 1 Report

Thank you for the opportunity to review this manuscript. The main aim of this study was to evaluate gender-specific associations between HL and PL in Croatian adolescents.

I have some suggestions to refine this manuscript.

The introduction could be implemented. In particular, I think it's important to underline the importance to support physical activity in adoscents population througt different strategy. In particular i't should be stressed that physical activity education begins as a child. How? With different way, also throught the game. Here some references "La Torre G, Mannocci A, Saulle R, Sinopoli A, D'Egidio V, Sestili C, Manfuso R, Masala D. [GiochiAMO! The protocol of a school based intervention for the promotion of physical activity and nutrition among children]. Clin Ter. 2016 Sep-Oct;167(5):152-155. Italian. doi: 10.7417/CT.2016.1947. PMID: 27845482"."Neil-Sztramko SE, Caldwell H, Dobbins M. School-based physical activity programs for promoting physical activity and fitness in children and adolescents aged 6 to 18. Cochrane Database Syst Rev. 2021 Sep 23;9(9):CD007651. doi: 10.1002/14651858.CD007651.pub3. PMID: 34555181; PMCID: PMC8459921".

Methods are well structured.

Conclusione are clear.

Author Response

Thank you for the opportunity to review this manuscript. The main aim of this study was to evaluate gender-specific associations between HL and PL in Croatian adolescents.

I have some suggestions to refine this manuscript.

RESPONSE: Thank you for your support. We hope that we amended the manuscript sufficiently.

The Introduction could be implemented. In particular, I think it's important to underline the importance to support physical activity in adoscents population througt different strategy. In particular i't should be stressed that physical activity education begins as a child. How? With different way, also throught the game.
Here some references
"La Torre G, Mannocci A, Saulle R, Sinopoli A, D'Egidio V, Sestili C, Manfuso R, Masala D. [GiochiAMO! The protocol of a school based intervention for the promotion of physical activity and nutrition among children]. Clin Ter. 2016 Sep-Oct;167(5):152-155. Italian. doi: 10.7417/CT.2016.1947. PMID: 27845482".
"Neil-Sztramko SE, Caldwell H, Dobbins M. School-based physical activity programs for promoting physical activity and fitness in children and adolescents aged 6 to 18. Cochrane Database Syst Rev. 2021 Sep 23;9(9):CD007651. doi: 10.1002/14651858.CD007651.pub3. PMID: 34555181; PMCID: PMC8459921".

RESPONSE: The Introduction has been corrected. Text now reads: "Indeed, education regarding physical activity and healthy nutrition begins from a very young age (i.e., as a child) in the perspective of active games [2]. This emphasizes the importance of early education and acquisition of healthy habits, as it relates to improved physical fitness and cardio-metabolic health [3]". Please see the first paragraph of the Introduction. (Note that references 2 & 3 are those you have suggested in your comment)

Methods are well structured.

RESPONSE: Thank you!

Conclusione are clear.

RESPONSE: Thank you for your support!

Reviewer 2 Report

First of all, I would like to congratulate the authors for launching this novel study, with a very topical theme. However, I think it needs a different orientation. 

The results as they themselves point out are not as expected and in this sense I have several comments that may help the authors to improve the manuscript. 

Major comments: 

One of the main limitations they do not mention is that the PLAY self questionnaire is validated for children between 8 and 14 years old: https://www.researchgate.net/publication/347875130_Psychometric_properties_and_construct_validity_of_PLAYself_a_self-reported_measure_of_physical_literacy_for_children_and_youth.

However, they use it in an older population whose physical, psychological and psychological changes with respect to the previous age are very evident. 

In this sense, I have doubts as to whether this is the most appropriate measure, which is why the results may not be as expected. 

Furthermore, the authors point out in the results that the sample comes from a medical school, which is not specified in the respective section on participants or in the limitations of the study. 

In this sense, if the sample is as specific as they highlight, perhaps the study should not be called "Are health literacy and physical literacy independent concepts? 2 Gender-stratified preliminary study in Croatian adolescents", as it does not refer to the general Croatian adolescent population, but to a specific sample of adolescents. It is recommended to the authors to orient the manuscript in this direction, as it would gain in meaning their obtained results. 

Minor comments: 

The wording of the method is poor. 

The study design is not described. What type of study is it? Please specify. In this sense, figure 1 does not correspond to the design, it corresponds to the measures. 

Pearson's product moment correlation? What does this mean?

In the results section, conjectures are made that are more appropriate for discussion, it is recommended that the authors limit themselves to stating their results in that section (Apart from logical significant correlations between anthropometric/body build variables...; In brief, boys and girls differed significantly in anthropometric-body composition measures as expected).

In the conclusions, there are some limitations and guidelines for the future which should come before them (324-333).

Author Response

Reviewer 2

First of all, I would like to congratulate the authors for launching this novel study, with a very topical theme. However, I think it needs a different orientation.

The results as they themselves point out are not as expected and in this sense I have several comments that may help the authors to improve the manuscript.

RESPONSE: Thank you for your support! We tried to amend the manuscript according to your comments and suggestions and hope it is now improved.

Major comments:

One of the main limitations they do not mention is that the PLAY self questionnaire is validated for children between 8 and 14 years old: https://www.researchgate.net/publication/347875130_Psychometric_properties_and_construct_validity_of_PLAYself_a_self-reported_measure_of_physical_literacy_for_children_and_youth.

However, they use it in an older population whose physical, psychological and psychological changes with respect to the previous age are very evident.

In this sense, I have doubts as to whether this is the most appropriate measure, which is why the results may not be as expected.

RESPONSE: We partly agree with you. PLAYself is evaluating the self-description of one's perceived PL. Thus, we considered these questions appropriate even for older children, as they perceive their PL according to their peers and not younger/older children. Thus, we considered that PLAYself could also give good feedback about self-described PL even among older children and adolescents, as we had in our study. Indeed, several recent studies have confirmed the reliability and validity of the PLAYself questionnaire in Croatian adolescents aged 14-19 years, as in this study.
Please see:
https://www.mdpi.com/2227-9067/9/6/796; https://pubmed.ncbi.nlm.nih.gov/35626930/

However, we are aware that PLAYself may not be sufficient for assessing a whole PL level, as it does not include all PL domains but only cognitive and affective ones. Thus, we are aware that in future studies, it should be better to have also other PL tools which would make the results more straightforward and precise. However, we consider that this manuscript could be of great importance for initiating further research regarding these essential health-related concepts.

This has been added in the Conclusions section: “Also, the tool used for assessing PL (i.e., PLAYself) is probably not sufficiently appropriate for assessing PL in this sample, nor for assessing all domains of PL as it includes only cognitive and affective ones. Therefore, more appropriate and wider range of PL tools that include all PL domains should be used in future studies, to assess PL in more detail”.

Furthermore, the authors point out in the results that the sample comes from a medical school, which is not specified in the respective section on participants or in the limitations of the study.

In this sense, if the sample is as specific as they highlight, perhaps the study should not be called "Are health literacy and physical literacy independent concepts? 2 Gender-stratified preliminary study in Croatian adolescents", as it does not refer to the general Croatian adolescent population, but to a specific sample of adolescents. It is recommended to the authors to orient the manuscript in this direction, as it would gain in meaning their obtained results.

RESPONSE: We agree entirely with you (we studied only high-school students from only one region in Croatia, and indeed significant proportion of participants were  from medical). Thus, we changed the title of the manuscript to: "Are health literacy and physical literacy independent concepts? Gender-stratified preliminary study in high school students from southern Croatia"

Also, it has been mentioned in the Methods and Limitations part of the manuscript. Text now reads: "The study was conducted on 253 adolescents (16.9±1.4 years of age, 181 females) attending high schools in Split-Dalmatia County, in southern Croatia".

and

" Another limitation was the homogeneity of the studied group, as we included exclusively high school students from only one region in Croatia, and significant proportion of the studied sample were students of medical high-schools, which limits the possibility of drawing broader conclusions regarding this topic.”

Also, it has been mentioned in the Conclusion, and text reads: “The specificity of the studied sample and study period could explain the lack of differences in HL between male and female adolescents. Precisely, this study included high-school adolescents from only one region and limited number of schools, and testing was done in the period of COVID-19 (when awareness of the health-related is-sues generally increased). Thus, future studies should investigate students from other regions and ages to determine a more precise state of the examined issue. Also, the tool used for assessing PL (i.e., PLAYself) is probably not the best option for assessing PL in this sample, nor for assessing all domains of PL as it includes only cognitive and affective ones. Therefore, more appropriate and wider range of PL tools that include all PL domains should be used in future studies, to assess PL in more details.” (please see Conclusion section for more details)

Minor comments:

The wording of the method is poor.

The study design is not described. What type of study is it? Please specify. In this sense, figure 1 does not correspond to the design, it corresponds to the measures.

RESPONSE: Thank you for this valuable comment. We added the type of the study, and the text now reads: "This cross-sectional study included evaluation of HL, PL, and anthropometric/body-build indices".

Also, we changed the title of the Figure 1 to Study variables and measurement and the description of Figure 1: "Study variables, measurement and idea are presented in Figure 1".

Please see Methods section.

Pearson's product moment correlation? What does this mean?

RESPONSE: The Pearson product-moment correlation coefficient (Pearson's correlation, for short) is a measure of the strength and direction of association that exists between two variables measured on at least an interval scale. (see for example here: https://statistics.laerd.com/spss-tutorials ). We tried to briefly explain it in the text now, and text reads: „Pearson’s product moment correlation coefficient, as a measure correlation between two normally distributed variables, was calculated to evaluate association be-tween pairs of variables, and this was done for total sample, and gender-stratified.“ (please see subsection Statistical analyses). Thank you.

In the results section, conjectures are made that are more appropriate for discussion, it is recommended that the authors limit themselves to stating their results in that section (Apart from logical significant correlations between anthropometric/body build variables...; In brief, boys and girls differed significantly in anthropometric-body composition measures as expected).

RESPONSE: Thank you for noticing it. By all means, this “discussion moments” were not appropriate for Results section. In this version of the manuscript we tried to stick to presenting results, and all parts of the text where some kind of discussion was implicated are now avoided. Please see Results section for details.

In the conclusions, there are some limitations and guidelines for the future which should come before them (324-333).

RESPONSE: We added limitations and guidelines that we consider also important. Text now reads: “Also, the tool used for assessing PL (i.e., PLAYself) is probably not sufficiently appropriate for assessing PL in this sample, nor for assessing all domains of PL as it includes only cognitive and affective ones. Therefore, more appropriate and wider range of PL tools that include all PL domains should be used in future studies, to assess PL in more detail”. Please see Conclusions section.

Thank you once again for recognizing the importance of our research and for the opportunity to amend our manuscript.

Staying at your disposal.

Authors

Round 2

Reviewer 2 Report

The authors have responded to most of the comments suggested, however, I have some antotations. 

Regarding the change of title, I think it is not enough, as they specify "secondary" as a very general population yet, as this does not include students in the health area. I think this is very important as it may be one of the justifications for the results of your study. 

The design is not included in the corresponding section where they indicate it as a title (line 91), it is included in the following section. 

The conclusions are still unclear, they are a mixture of results, limitations and future directions, they are too extensive. 

Author Response

Reviewer 2

The authors have responded to most of the comments suggested, however, I have some antotations. 

RESPONSE: Thank you for recognizing the improvements. Please see how we amended the manuscript according to your additional comments.

Regarding the change of title, I think it is not enough, as they specify "secondary" as a very general population yet, as this does not include students in the health area. I think this is very important as it may be one of the justifications for the results of your study. 

RESPONSE: Following your suggestion the title is amended and now reads: „Are health literacy and physical literacy independent concepts? Gender-stratified preliminary study in high school health-students from Croatia“

The design is not included in the corresponding section where they indicate it as a title (line 91), it is included in the following section. 

RESPONSE: Thank you for noticing it. It is amended accordinlgy. The first sentence of the „Participants and design“ subsection now contains details about study design. Text reads: „In this cross-sectional study participants were 253 adolescents (16.9±1.4 years of age, 181 females) attending high schools in Split-Dalmatia County in southern Croatia“

The conclusions are still unclear, they are a mixture of results, limitations and future directions, they are too extensive. 

RESPONSE: Thank you. The Conclusion section is systematically rewritten and shortened, and we hope that it is now more „focused“. Text reads:

„As the results showed weak associations between HL and PL, our research high-lights the necessity for the separate evaluation of each of these important abilities in adolescence. Indeed, despite both HL and PL being related to health behaviors and outcomes, they are evidently different concepts and should be assessed separately.

The tool used for assessing PL (i.e., PLAYself) is probably not the best option for assessing PL in high-school adolescents, nor for assessing all domains of PL as it in-cludes only cognitive and affective ones. Therefore, in future studies, all aspects of PL should be observed to objectively evaluate PL itself and to establish the associations that may exist among PL, HL, and overall health-related status.

This study included anthropometric/body composition indices as the only varia-bles of health status. Therefore, in future investigations other indices of health are needed in order to objectively evaluate the eventual connection that may exist among HL, PL, and health status.“

Thank you once again!